# Experiences of general practice of children with complex and palliative care needs and their families: a qualitative study

Sarah Mitchell  ,[1,2] Stephanie Harding,[2] Mohini Samani,[3] Anne-Marie Slowther,[2] Jane Coad,[4] Jeremy Dale  [2]

[1]Oncology and Metabolism, The University of Sheffield, Sheffield, UK
[2]Warwick Medical School, University of Warwick, Coventry, UK
[3]NIHR CRN West Midlands Young Person's Steering Group, Stafford, UK
[4]Faculty of Medicine and Health Sciences, University of Nottingham, Nottingham, UK

**Correspondence to**
Dr Sarah Mitchell;
s.j.mitchell@sheffield.ac.uk

## ABSTRACT

**Objectives** To investigate the views and experiences of general practice of children with life-limiting and life-threatening conditions, and their family members, through secondary analysis of a qualitative serial interview study. Thematic analysis was conducted on all interview data relating to experiences of primary care.

**Setting** West Midlands, UK.

**Participants** A total of 31 participants (10 children with life-limiting and life-threatening conditions and 21 family members) from 14 families.

**Study design and setting** Secondary thematic analysis of qualitative interview data from a study carried out in the West Midlands, UK.

**Method** 41 serial interviews with 31 participants from 14 families: 10 children aged 5–18 years with life-limiting and life-threatening conditions, and 21 of their family members.

**Results** Three key themes emerged: (1) poor experiences of general practice cause children and families to feel isolated, (2) children and families value support from general practice, and (3) there are practical ways through which general practice has the potential to provide important aspects of care. Children and families reported benefits from fostering their relationship with their general practice in order to access important aspects of care, including the assessment and management of acute illness, chronic disease and medication reviews, and holistic support.

**Conclusion** Children with life-limiting and life-threatening conditions and their families value the involvement of general practice in the care, alongside their paediatric specialists. Ways of developing and providing such support as part of an integrated system of care need to be developed.

## INTRODUCTION

The numbers of children and young people (hereafter referred to as 'children') aged 0–19 years living with life-limiting conditions (for which there is no reasonable hope of cure) and life-threatening conditions (those for which curative treatment is feasible but may fail) is rising rapidly.[1–3] As medical treatments

### Strengths and limitations of this study

► Little is known about the role of general practice in the care of children with life-limiting and life-threatening conditions and their families, including aspects of palliative care.

► This study provides insights into the support and care from general practice that children and their family members find most helpful from their perspective, including their strategies for accessing such care.

► The study used longitudinal qualitative interviews, which provided the benefit of developing rapport and gaining in-depth insights into the experiences of children and their families.

► There was diversity among the children and families interviewed in terms of their conditions, family circumstances, ethnic background and geographical location (inner city, urban and rural).

► This study is a subanalysis of a wider study that did not specifically focus on experiences of general practice, and as such, the dataset is limited, and there is more work to do to understand the specific views of children.

and technology advance, these children are living with increasingly complex conditions. Their health is often fragile, with sudden and unanticipated deterioration leading to admission to hospital and intensive care. This has a huge impact on family life and presents a number of challenges for healthcare services, including the management of complex clinical issues and the provision of adequate care and support, including palliative care, as close to home as possible.[4]

The involvement of general practitioners (GPs) and primary care teams in the provision of healthcare to children with life-limiting and life-threatening conditions can be variable. These children, like all others, have primary care needs in addition to the specialist needs associated with their

underlying conditions. Those with life-limiting and life-threatening conditions may also have associated palliative care needs.[2] GPs have a key role in the delivery of 'core' palliative care to all patients who need it. For children, this would be alongside community nursing and specialist paediatric palliative care colleagues. However, GPs have expressed significant concerns about this role in palliative care for children, including a lack of time to be involved, lack of specialist knowledge (particularly with rare conditions) and understanding of their role alongside paediatric colleagues.[5 6] However, GPs have the potential to provide many vital aspects of palliative care, including the coordination of care from community and specialist teams, provision of prescriptions, holistic support for family members (including during bereavement) and support when children transition from paediatric to adult palliative care services.[7] Furthermore, frequent appointments with a GP and continuity of GP care for children with life-limiting and life-threatening conditions are associated with less frequent use of urgent and emergency secondary care services.[8]

This study aimed to explore the role of general practice in the care of children with life-limiting and life-threatening conditions and their families, from their perspective, and consider how this might be developed to more effectively support the delivery of care to these children in the future.

## METHODS

This study is a secondary analysis of data relating to primary care collected during a qualitative serial interview study carried out from October 2016 to November 2017 in the West Midlands, UK.[9] The overall aim of the study was to examine the delivery of healthcare, including palliative care, for children with life-limiting and life-threatening conditions and their families, using a realist approach to provide understanding into how palliative care is delivered most effectively, and when, leading to policy-relevant recommendations. Longitudinal interviews provided further benefits, including opportunities for rapport building and observation of changing needs and experiences of healthcare over time.[10]

Children and family members were provided with study information leaflets either by their clinical teams, or via leaflets and posters in public areas including waiting rooms of hospital outpatient departments. In order to avoid possible coercion, clinical teams did not actively recruit children and families to the study. Children and family members were recruited after either directly approaching the research team via email, text or telephone using the details provided in the study information resources, or after expressing an interest in participation and providing permission to be contacted. Each child and family member participated in up to three interviews over the 13-month period, either individually or as a family group. Qualitative research methods were most appropriate in order to elicit and explore the views and

perceptions of children and their families. The interview process, including whether children wished to be interviewed alone or with their family members on each occasion, the timing and intervals between interviews, were individually considered according to the needs of each child and family. The interviews were open and conversational, and advanced communication skills, including responding to cues and the use of silence, were employed. This allowed exploration of potentially sensitive issues, including experiences of palliative care as both a broad approach to care, and when the child and family had contact with specialist paediatric palliative care services.

Consent procedures were designed with the aim of obtaining written and/or verbal consent and agreement from every individual for every interview. For children under the age of 16 years, written consent was obtained from the parent and then verbal or written agreement obtained from the child. An agreement-to-participate form was completed by the child if they chose to do so, in order to respect their autonomy in so far as was possible.

Recordings of interviews were transcribed verbatim, and anonymised transcripts uploaded into NVivo for data management. All data related to experiences of general practice were coded by SM and formed the dataset for this secondary analysis. Two researchers (SH and SM) independently coded the data using an inductive approach to thematic analysis as described by Braun and Clarke.[11 12] This started with a process of familiarisation, reading and rereading the transcripts, with reflection and note-taking. A descriptive code was applied to every item of data and then emerging codes and concepts were discussed between the researchers weekly throughout the data analysis process, allowing for the development of the themes and decreasing lone researcher bias.[13]

### Patient and public involvement

Patient and public involvement (PPI) was an integral part of the wider study and informed every stage of the research. A group of PPI representatives was recruited from three existing advisory groups (based at Birmingham Children's Hospital, Acorns Children's Hospice and the National Institute for Health Research Clinical Research Network in the West Midlands). The PPI representatives were aged between 9 and 25 years, and included young people with life-limiting conditions and siblings. The group were consulted at all stages of the research, from the study design and objectives to the format of interviews for children, and dissemination activities. A PPI representative (MS) volunteered to support this subanalysis, providing feedback on the emerging themes and supporting dissemination.

### Findings

#### Study sample

A total of 14 families were recruited to the study, all registered with different GP practices, urban, inner city and rural. Ten children (aged 5–18 years) and 21 family members took part in the interviews. Children took

part in the interviews either with their family members or alone, according to their preference on the day of the interview. Four children were unable to take part either because they could not communicate verbally, or because they were too unwell. Forty-one interviews were conducted in total. All of the children received care from a local community children's nursing team, and six received care from a specialist paediatric palliative care team or children's hospice service. The study population is summarised in table 1.

## Qualitative findings

The children and their family members described a range of experiences of general practice both positive and negative. They did not tend to associate GPs with providing aspects of palliative care; all of the children and families who received palliative care services considered and conceptualised 'palliative care' as distinct and separate from other services. Three overarching themes relating to general practice were identified:

### Theme 1: poor experiences of general practice cause children and families to feel isolated

For many, there was a lack of contact with GPs and practice teams. Several families had negative experiences and chose to avoid the GP surgery completely due to these experiences, tending to contact their specialist teams for help instead. Most described their GPs as being of little or 'no help at all' (M002). Many family members described a 'fight' to access the support from their GP practice that they required:

> I haven't got the power, the brain power, to deal with the doctors' surgery. Literally, it's just like I can't. The doctors' surgery is a no go. I avoid it. It will be my last call, if I have to. (M005)

Negative experiences led to a loss of trust in the GP that family members felt unable to reconcile. In these circumstances, there was a sense of unfortunate loss of an important source of support. In one example, the GP was considered responsible for missing a significant diagnosis (cancer) in the child:

> he didn't realise. You know, when you're talking life and death for your child, you can't make mistakes like that. But it was… and most GPs don't see things like this. It's because… they say it's rare, it doesn't feel rare to us in here [the hospital]. (F007)

Some family members had tried to access support from their GP for their own distress related to their child's condition. They sought a listening ear and validation of their distress, but perceived that the GP had limited time to hear and understand, and would focus on providing a medical solution such as an antidepressant tablet. This led to further frustration:

> [the GP] gives me tablets all the time. What if the palpitations are something else? I avoid them now.

They are so busy, over-stretched. I think the empathy has gone. (M013)

Another frustration arose on occasions where GPs declined to prescribe regular medications for the children. Family members, who were experts in the management of their children's conditions, found it difficult to understand the rationale for this:

> Everything is like, oh, I can't prescribe this because it's unlicensed, or it's un-this, and simple things, [drug names for emollients], he wouldn't do it. (M005)

Family members recognised that the complexity of their child's condition, and their extensive specialist care, as potential barriers to GPs becoming more involved, with GPs lacking confidence around certain aspects of care, such as prescribing or treating acute medical problems.

> …an awful battle just to try and get antibiotics for her for a urine infection, it was terrible. (M003)

> As soon as [child]'s [life-limiting condition] problems started they just pulled back from everything. Even her basic [medications] and whatever she needed, it all just stopped. (M002)

Even with minor illness, families were advised to access hospital care, which they did not always agree was appropriate:

> …you're bringing this person up to the [children's hospital] who doesn't really need to be at the [children's hospital] and affecting everybody else because the GP won't come out. (M011)

### Theme 2: children and families value support from general practice

Some families described positive experiences of general practice, considering their GP to be 'realistically the core of it all' (M003), even when hospital-based, specialist teams mainly provided their care. Some had known a particular GP for many years, and highly valued this continuity, with the GP 'knowing our family' (M006). For one family, it was the GP who 'finally realised' (F004) that they were in need of extra support and arranged an onward referral to community children's nursing and palliative care services. Being able to consider the GP as an important member of their healthcare team, through, for example, their attendance when the child had an acute illness, so that they could avoid a trip to hospital, made a significant difference to families where this was the case:

> They're good at coming out if he has a chest infection, they do come out, yeah. Because at the end of the day I've got a child that's on a sats monitor, and ventilator, and now a humidified circuit, and he's not well. (M014)

Children and their family members all expressed a desire for more support from the GP practice. They wished for a proactive offer of support, with recognition of their holistic care needs, through 'a general check

**Table 1** Summary of study population

| Family | Participants and identifier | Child's age at recruitment | Type of condition | Male or female | GP practice area | Palliative care service involved | Total number of interviews | Interview details |
|---|---|---|---|---|---|---|---|---|
| 1 | Child (C001)<br>Mother (M001)<br>Father (F001) | 5 | Cancer | M | Urban | No | 4 | 1: mother<br>2: mother and child individual interviews<br>3: family group |
| 2 | Child (C002)<br>Mother (M002)<br>Brother (B002) | 17 | Congenital | F | Inner city | Yes | 3 | 3× family group interviews all including child |
| 3 | Mother (M003)<br>Father (F003) | 8 | Congenital | F | Inner city | Yes | 3 | 1: mother<br>2: both parents<br>3: mother |
| 4 | Father (F004) | 8 | Congenital | F | Inner city | Yes | 1 | 1: father |
| 5 | Child (C005)<br>Mother (M005) | 6 | Congenital | M | Urban | No | 3 | 3× family group interviews all including child |
| 6 | Mother (M006) | 18 | Congenital | M | Inner city | No | 3 | 3× interviews with mother |
| 7 | Child (C007)<br>Mother (M007)<br>Father (F007) | 7 | Cancer | M | Urban | No | 5 | 1: family group<br>2: parents and child individually<br>3: mother and child individually |
| 8 | Child (C008)<br>Mother (M008)<br>Brother (B008) | 5 | Congenital | M | Inner city | No | 4 | 1: family group<br>2: mother and child individual interviews<br>3: family group |
| 9 | Child (C009)<br>Mother (M009)<br>Father (F009) | 11 | Cancer | F | Inner city | No | 3 | 3× family group interviews all including child |
| 10 | Mother (M010) | 5 | Congenital | M | Urban | Yes | 1 | 1: mother (child present but too unwell to contribute) |
| 11 | Child (C011)<br>Mother (M011)<br>Stepfather (F011) | 17 | Congenital | F | Urban | No | 4 | 1: family group<br>2: family group<br>3: family and child individual interviews |
| 12 | Child (C012)<br>Mother (M012) | 14 | Cancer | M | Urban | No | 1 | 1: family group |
| 13 | Child (C013)<br>Mother (M013) | 14 | Cancer | M | Urban | Yes | 3 | 1: mother<br>2: mother and child interviews |
| 14 | Child (C014)<br>Mother (M014) | 10 | Congenital | M | Rural | Yes | 3 | 3× family group interviews (including child) |

GP, general practitioner.

on the normal everyday' (M005), for example through a phone call. This was considered particularly important when there was a long time between hospital appointments. One child described this time as a time when 'no-one knows what is going on' and specifically regarded the GP or a member of the practice team as someone who could 'check-in' with him (C013).

> …that would be perfect if there was a somebody there that every six months just said, right, [name], how is everything going in the home, how is everything with. all your meds, have you got any problems, is there anything I can do for you, can I phone anyone, do you need anything, just a someone. (M013)

### Theme 3: practical ways through which general practice could provide improved care

Family members described the aspects of care that they wanted general practice teams to provide, and made efforts to 'consciously foster' (M006) their relationships with GPs in order to access this care. Opportunities existed in chronic disease reviews, appointments for vaccinations and sick notes:

> I have always been quite proactive in taking my children to the GP not only when they're sick. I would take them [children] along for flu jabs and things like that, just so that they knew what they're like, so that when I take them when they're sick they know the difference. (M006)

Interventions that children and family members felt would be helpful for general practice teams to offer included blood tests, so that they did not have to travel long distances to hospital. Where repeat prescriptions were a particular cause of difficulty, members of the wider team, including practice pharmacists, had been instrumental in improving the situation through review of discharge summaries and updating a regular prescription:

> As soon as they get a letter from the hospital to say her meds changed they have someone ring me and we go through her meds. (M011)

Transition to adult services was a cause of concern for many families. They considered this a time when it would be helpful for support from general practice to 'really kick in' (M003), particularly because organisation of healthcare services for adults differed significantly to those for children:

> I didn't realise that in adult care, a lot of help is from the GP and the community nurse, but we didn't know that because it's different at the [children's hospital], and when we went to the [hospital] this consultant, and neuro, said, they [primary care] have to [prescribe the medicines] but they won't do it. (M002)

The need for a consistent and familiar trusted contact within the surgery for children who may want to attend appointments alone was highlighted:

> how the hell can I transition my son when the GP has never ever called … I'm trying to explain to [child], you will be old enough to go to the GP on your own. That's immense for a normal kid, but a kid with all [his conditions]… he won't talk about all of his…I want him to, because I think it would be a good idea if there was a particular person in the GP's. (M013)

## DISCUSSION
### Summary
This study provides valuable insights into the highly variable experiences of general practice of children with life-limiting and life-threatening conditions and their family members. For some, there were some positive examples of general practice involvement in the child's care. Generally, the relationship was experienced as unsatisfactory. This could result in a loss of trust, avoidance, or years of battling to obtain the care and support required. Both children and family members expressed a wish to receive more support from their GP practice, and importantly had developed strategies in order to access the support that they found most helpful.

### Strengths and limitations
A strength of this study is that it considered the experiences and perceptions of general practice from the perspective of children and their families. Family members tended to talk more about experiences of general practice, which is likely to reflect their key involvement in liaising and negotiating services from GPs. This was a subanalysis of a wider study that did not specifically focus on experiences of general practice, and as such, the dataset is limited. The views of children and family members specifically in relation to general practice are worthy of future research, and future studies with a focus on community healthcare, including primary care, are recommended.

The study population was relatively small, but it was diverse in terms of the children's condition and age. The participants were registered with 14 different practices, in a broad range of different settings. In-depth serial interviews, through which a relationship with children and their families could develop, led to rich, detailed data elicited around many different aspects of care. PPI was gained at several stages of the research, including the development of themes during the analysis to ensure the research remained representative and sensitive.

There is more research to be done to understand the experiences of younger children, young people at transition and those with non-verbal communication. Further research into the role and response of primary care beyond general practice, including dentists, opticians and out-of-hours services for children with life-limiting and life-threatening conditions, would also be valuable.

### Comparison with existing literature
Care for children with complex and palliative care needs in general practice is an under-researched area

of practice. The children and families who participated in this study described the particular aspects of care that would be helpful for primary care to provide, and their experiences of how such care could be delivered. The findings are in keeping with the current, limited evidence base, which has explored the experiences of GPs in palliative care of children with cancer,[7 8 14] and the perspectives of bereaved parents.[15] Continuity of care in general practice, effective communication with hospital-based, specialist care and bereavement support are areas described repeatedly as needing improvement. Furthermore, the association between more regular contact with GPs and less frequent use of urgent hospital care warrants further investigation in order to understand the factors that result in this outcome for some children, and the details of their relationships with general practice.[6] A particularly important area for consideration is transition from paediatric to adult services. This is described as an 'overwhelming process' by some parents, with a sense of loss of paediatric services. Continuity of care and support from the GP practice in this scenario has many potential benefits for both parents and children with complex and palliative care needs but is often overlooked.[16]

## Implications for clinical practice and research

Recent research shows that the number of children with life-limiting or life-threatening conditions increased in England, that it has risen from 32 975 in 2001/2002 to 86 625 in 2017/2018.[3] This is double the total GP headcount in England, which in March 2017 was 41 891 (33 921 whole time equivalent GPs).[17] The role and response of general practice in the healthcare of these children and their families needs greater consideration.

The findings of this study suggest a need for a more systematic proactive approach, such as through prescribing and chronic disease reviews, to provide continuity of care and to build valuable therapeutic relationships with children with life-limiting and life-threatening conditions and their families. GPs and practice team members have the skills and ability to address the physical, psychological, social and spiritual aspects of care, through these relationships and as part of a wider multidisciplinary team. These aspects of care may not necessarily be labelled as 'palliative' care, but they provide an important foundation and are integral to the delivery of holistic, palliative care to children, which by the nature of children's conditions, may be needed over a prolonged time period. GPs could play an increasingly important role in the identification of palliative care need and onward referral to specialist services, including communication with specialists in paediatric palliative care and children's hospice services. Having a named GP within the practice, who becomes the key point of contact for the family and the specialist team, becomes familiar with the needs of the child and family, and undertakes a regular review in which a care plan is agreed, would be of value.

Developing the skills, knowledge, expertise and confidence levels of GPs and other members of the practice team, perhaps those with a particular interest in paediatrics, or management of complexity, to assess acute illness in children with life-limiting and life-threatening conditions would also be of value. In theory, general practice is well placed to work with these children and families, however general practice is under-resourced and may not currently be in a position to prioritise this care.

At a healthcare system level, the need for collaborative partnership working between primary care clinicians and specialist paediatric colleagues in order to improve the experiences of care of children with life-limiting and life-threatening conditions and their families is important. Interventions, such as clear and personalised shared care plans, and shared electronic medical records, would be helpful. National professional bodies, including the Royal College of General Practitioners and the Royal College of Paediatrics and Child Health in the UK, have a key role to play in driving forward innovations and effective interventions to facilitate collaborative care. National and local commissioning strategies require focus on this population of children, and should support the delivery of their care in both hospital and community settings.

## CONCLUSION

The findings from this study suggest that general practice should be playing a more important role in the provision of holistic, family-centred healthcare to children with life-limiting and life-threatening conditions and their family members. Children and families described multiple benefits associated with consciously fostering their relationship with general practice clinicians in order to access important aspects of care, including continuity. There are opportunities to engage with this through chronic disease and medication reviews. Clinicians in general practice could develop their skills and expertise in this area to provide assessment and management of acute illness, and long-term holistic support, in close partnership with family members and paediatric specialists.

**Acknowledgements** The authors would like to acknowledge all of the children and families who took part in this study, and all of the patient and public involvement representatives from Birmingham Children's Hospital, Acorns Children's Hospice and the NIHR CRN West Midlands Young Person's Steering Group who contributed to all aspects of the study.

**Contributors** SM, JD, A-MS and JC designed the study. SM conducted the interviews. SM and SH analysed the data, with input from MS as a representative of the PPI advisory group. JC, A-MS and JD revised the article critically for clarity and intellectual content. All authors have approved this version for submission.

**Funding** This study was funded by a National Institute for Health Research Doctoral Research Fellowship (DRF-2014-07-065).

**Disclaimer** This article presents independent research funded in part by the National Institute for Health Research (NIHR). The views expressed are those of the authors and not necessarily those of the NHS, the NIHR or the Department of Health.

**Competing interests** None declared.

**Patient and public involvement** Patients and/or the public were involved in the design, or conduct, or reporting, or dissemination plans of this research. Refer to the Methods section for further details.

**Patient consent for publication** Not required.

**Ethics approval** Ethical approval was granted by the UK Health Research Authority on 14th September 2016 (IRAS ID: 196816, REC reference: 16/WM/0272, Sponsor: University of Warwick). The data were anonymised and personal identifiers removed to protect confidentiality, as described in the Methods section.

**Provenance and peer review** Not commissioned; externally peer reviewed.

**Data availability statement** Data are available upon reasonable request. Individual deidentified participant data that underlie the results published in this article will be made available from 3 months until 3 years following the article publication to investigators with a methodologically sound proposal and whose proposed use of the data has been approved by an independent review committee identified for this purpose, to achieve the aims in the approved proposal. Proposals can be submitted to the corresponding author.

**ORCID iDs**
Sarah Mitchell http://orcid.org/0000-0002-1477-7860
Jeremy Dale http://orcid.org/0000-0001-9256-3553

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
