## [Reviewer comments · BMJ Open]

ARTICLE DETAILS

TITLE (PROVISIONAL)	Experiences of General Practice of children with complex and palliative care needs and their families: a qualitative study
AUTHORS	Mitchell, Sarah; Harding, Stephanie; Samani, Mohini; Slowther, Anne-Marie; Coad, Jane; Dale, Jeremy

VERSION 1 – REVIEW

REVIEWER	Maha Atout Philadelphia university / Jordan
REVIEW RETURNED	04-Jul-2020

GENERAL COMMENTS	Thank you for providing me the opportunity to review this study. Overall, it is an important study to shed light on an essential aspect of paediatric palliative care, which is an under-researched area of practice. However, I have some suggestions: • Briefly, could you define the serial qualitative interview method and explain the aim of its adoption in this study?• Could you review Table 1, please? The number of interviews is thirty-five. The number of children who participated in the interviews is probably less than ten.• Page 5, line 22: Could you provide some contextual information about the Patient and Public involvement (PPI), please? I did not grasp whether PPI is a team explicitly assigned for this study or a team that has already worked in the hospital? Regards
---

REVIEWER	Jane Evered University of Wisconsin-Madison
REVIEW RETURNED	27-Aug-2020

GENERAL COMMENTS	Thank you so much for the opportunity to read this qualitative study highlighting the often not well-understood experience of families managing life-limiting/life-threatening illnesses. The data collected is richly informative and the significance of the topic is well presented. The in-depth serial interviews are a major methodological strength, a source of immense insight into the care needs of these children. Background: The background contextualizes the aims of the study well. To strengthen the argument the author may consider why and how GPs provide aspects of palliative care and include more in-depth discussion of the types of palliative and other care. Methods: I recommend that the authors consider specifying their approach. It seems qualitative descriptive may align well, given the aims of the study and secondary nature of the data. It was not immediately clear to me what the authors mean by “explore the
---

	views of children and family members in relation to palliative care” – did the interviews focus on experiences of palliative care in general? Experiences with GPs who provided palliative care? Experiences with GPs in general? Even though this is a secondary analysis, I would like to see more detail about the recruitment and sampling for the primary study, as well as the interview process (ex. were children and families interviewed together?). It seems to me to be less relevant to include the findings and conclusions of the primary study in the table. I would suggest the authors consider focusing more on increasing the detail in the methods section. The authors might consider further elaborating on how rigor was maintained in this qualitative inquiry, as well as how the PPI team member was found and involved. Was there ethics approval? What was the consent process? Findings: In the findings, I would suggest referring to the group as a study sample instead of the study population. In the introduction to the findings, when refer to “palliative care” is this the palliative care that GP provides or that others provide? Overall, the quotes and examples are vivid and compelling. However, it struck me that most of the quotes presented are from family members – did family members speak more to the experience than children? In the first theme especially, I wonder if the authors might think about ways to integrate the quotes and analytic commentary a bit more to build the analysis further. In the second theme, I wonder whether further introduction might be added for some of the quotes. Overall, I’m left wanting to know more about how the themes relate. For a thematic analysis, it seems that most of the themes were more manifest rather than latent content and I wonder if there is potential for further interpretation and expansion. Discussion: While noted as a strength, the perspectives of children are largely missing from the findings. I encourage the authors to think about other limitations in the research study itself as the limitations noted are more future directions wonder if the comparison of the literature section could be expanded, as it is very similar to the introduction. In the implications, the authors might consider touching on palliative care aspects given the framing of the findings. Are there aspects of palliative care they recommend GPs provide beyond referral to specialist services? I’m grateful to the authors for this important work and appreciate the opportunity to review this manuscript.
--	---

REVIEWER	Sue Neilson University of Birmingham
REVIEW RETURNED	28-Aug-2020

GENERAL COMMENTS	This is a timely secondary analysis given the recognised increased prevalence of children with life limiting or life threatening conditions in England. Findings add to the knowledge base on the care of children with complex and palliative care needs and highlight implications for practice. The manuscript is clearly presented however there are 3 areas that require minor revision:  1. Pg 12: the end of the conclusion is missing in Box 1. 2. Pg 8 L19: punctuation error. 3. P9 L25 sentence starting 'Continuity of GP care..' : it is unclear if this should read 'hospital based specialist care and bereavement support'.
---

VERSION 1 – AUTHOR RESPONSE

Reviewer: 1

Briefly, could you define the serial qualitative interview method and explain the aim of its adoption in this study? I have added further detail to the methods section to explain and justify this method
Could you review Table 1, please? The number of interviews is thirty-five. The number of children who participated in the interviews is probably less than ten. I am sorry this is a mistake – I have oversimplified the table and in doing so have not provided sufficient detail about individual interviews and group interviews, both of which were conducted and counted in the total of 41. 10 children took part in interviews – some were as part of a group interview, where others were interviewed individually. I have updated the table and clarified in the text

Page 5, line 22: Could you provide some contextual information about the Patient and Public involvement (PPI), please? I did not grasp whether PPI is a team explicitly assigned for this study or a team that has already worked in the hospital? I have revised the PPI section on page 5 with further detail

Reviewer: 2

Background: The background contextualizes the aims of the study well. To strengthen the argument the author may consider why and how GPs provide aspects of palliative care and include more in-depth discussion of the types of palliative and other care. I have revised the introduction to provide more detail about the vital / core aspects of palliative care that GPs may be involve in providing, alongside community teams and specialist colleagues.

Methods: I recommend that the authors consider specifying their approach. It seems qualitative descriptive may align well, given the aims of the study and secondary nature of the data. It was not immediately clear to me what the authors mean by “explore the views of children and family members in relation to palliative care” – did the interviews focus on experiences of palliative care in general? Experiences with GPs who provided palliative care? Experiences with GPs in general? Even though this is a secondary analysis, I would like to see more detail about the recruitment and sampling for the primary study, as well as the interview process (ex. were children and families interviewed together?). It seems to me to be less relevant to include the findings and conclusions of the primary study in the table. I would suggest the authors consider focusing more on increasing the detail in the methods section. I have reviewed the methods section and added more detail. Having done so, I do not think there is a need for Box 1 anymore and so I have removed this.

The authors might consider further elaborating on how rigor was maintained in this qualitative inquiry. Detail is provided in the final methods paragraph on page 5 as well as how the PPI team member was found and involved. This detail has been added to the PPI section

Was there ethics approval? Details of the ethics approval are provided
What was the consent process? I have added this detail to the methods.

Findings: In the findings, I would suggest referring to the group as a study sample instead of the study population. I have changed the subtitle

In the introduction to the findings, when refer to “palliative care” is this the palliative care that GP provides or that others provide? This detail is provided in the first paragraph of the qualitative findings – the children and families associated the term “palliative care” with specialist services rather than a broad approach or something that was provided by the GP.

Overall, the quotes and examples are vivid and compelling. However, it struck me that most of the quotes presented are from family members – did family members speak more to the experience than

children? On reflection, the children did not contribute as much as their family members in relation to experiences of GP, so I have discussed this more as a limitation of the study

In the first theme especially, I wonder if the authors might think about ways to integrate the quotes and analytic commentary a bit more to build the analysis further. In the second theme, I wonder whether further introduction might be added for some of the quotes. Overall, I'm left wanting to know more about how the themes relate. For a thematic analysis, it seems that most of the themes were more manifest rather than latent content and I wonder if there is potential for further interpretation and expansion. I have reviewed this section and added more detail. Since this was a subanalysis, and experiences of GP were not the only subject of the overarching study, the depth of the analysis is a bit limited by the quality of the data. I have added this as a further limitation.

Discussion: While noted as a strength, the perspectives of children are largely missing from the findings. As above

I wonder if the comparison of the literature section could be expanded, as it is very similar to the introduction. I have not changed this as this really reflects the very limited literature in this area

In the implications, the authors might consider touching on palliative care aspects given the framing of the findings. Are there aspects of palliative care they recommend GPs provide beyond referral to specialist services? I have slightly edited this section of the discussion to describe the fact that important elements of care provided by GPs may not specifically be considered by GPs as "palliative", but in fact do provide an important foundation the delivery of specialist palliative care and end-of-life care.

Reviewer: 3

1. Pg 12: the end of the conclusion is missing in Box 1. Having added more detail to the methods I have now removed this box
2. Pg 8 L19: punctuation error - corrected
3. P9 L25 sentence starting 'Continuity of GP care..' : it is unclear if this should read 'hospital based specialist care and bereavement support'. I have rewritten this sentence for clarity.

I have corrected the formatting error noted.

The study was conducted as part of my PhD research, which was funded by a National Institute for Health Research Doctoral Research Fellowship (DRF-2014-07-065).

The enclosed manuscript has been read and approved by all authors. It is not under active consideration for publication elsewhere, has not been accepted for publication, nor has it been published in full or in part.

VERSION 2 – REVIEW

REVIEWER	Maha Atout Philadelphia university _Amman _Jordan
REVIEW RETURNED	24-Oct-2020

GENERAL COMMENTS	Well written study. Congratulations!
--------------------------------------

REVIEWER	Jane Evered University of Wisconsin-Madison, USA
REVIEW RETURNED	20-Oct-2020

GENERAL COMMENTS	Thank you for the opportunity to review this important manuscript and for your responses to the comments on the first submission. I appreciate the additions to the background that provide more detail
---

	about the aspects of the palliative care GPs are providing. Thank you for adding detail to the methods section about the primary care study. I'm wondering how the study team assured ethical research recruitment when having primary clinical teams approach children and families. The authors may consider adding this to the ethics section. I'd also urge the authors to consider adding slightly more detail about how the analytic process. The added information about PPI is very strong. You mention the quality of the data as a limitation that prevented deeper analysis in the findings. I'm curious in the limitations if you might explain more about what you mean by the quality of the data. The implications of the discussion are compelling and significant. Thank you for the opportunity to read this work.
--	---